

# A comparative evaluation of haematological and biochemical parameters of *Nemipterus randalli* and *Pagellus erythrinus* species living in Gökova Bay, Türkiye

Rifat Tezel[1], Ümit Acar[2] and Sercan Yapıcı[1]

[1] Faculty of Fisheries, Muğla Sıtkı Koçman University, Muğla, Türkiye
[2] Bayramiç Vocational School, Çanakkale Onsekiz Mart University, Çanakkale, Türkiye

Corresponding author
Sercan Yapıcı,
sercanyapici@mu.edu.tr

## ABSTRACT

This study aims to determine and compare the reference values of the haematological and biochemical blood parameters of two fish species collected from the Gökova Bay (Muğla, South-Western of Türkiye): the non-native and invasive Randall's threadfin bream, *Nemipterus randalli* and the native Common pandora, *Pagellus erythrinus*. Both species inhabit the same environment and compete for resources. Blood samples were collected from a total of 100 fish samples (50 *N. randalli* and 50 *P. erythrinus*) which were caught from a depth of 30 to 60 meters between February 2023 and July 2024. Therefore, sampling could be carried out in summer and winter seasons when the water temperature varies significantly. The results indicate that red blood cell (RBC), haemoglobin (Hb) and haematocrit (Hct) levels significantly increases in summer ($p < 0.05$). Statistical analyses revealed that all of the haematological parameters of *N. randalli* are higher than the *P. erythrinus* ($p < 0.05$). In terms of serum biochemical parameters, glucose (GLU), triglyceride (TRIG), cholesterol (CHOL), total protein (TP), aspartate aminotransferase (AST) and alkaline phosphatase (ALP) levels of two species had showed no significant difference in both summer and winter samples ($p > 0.05$). However, serum alanine aminotransferase (ALT) levels of *N. randalli* were statistically significant different compared to *P. erythrinus* in summer and winter samples ($p < 0.05$). Additionally, the highest lactate dehydrogenase (LDH) value was measured from *N. randalli* in summer sampling and this was found to be significantly different from other groups ($p < 0.05$). Multidimensional scaling (MDS) analysis revealed differences between the blood parameter results of *N. randalli* and *P. erythrinus* individuals. This study provides important data about the physiological adaptations of haematological and biochemical blood parameters of the two species to environmental factors. The similarity in haematological and biochemical parameters revealed the possibility that *N. randalli* could distribute to new habitats where *P. erythrinus* is distributed. This suggests a further potential distribution of the invasive *N. randalli* and the need for a careful monitoring.

## INTRODUCTION

The expansion of international commerce across recent centuries has led to the unintended transport of various species far beyond their native ranges (*Haubrock et al., 2024*). This human-driven dispersal has enabled numerous organisms to establish themselves in distant ecosystems. When these introduced species successfully colonize new territories, they can become invasive populations that pose significant ecological challenges (*Haubrock et al., 2024*). Their presence often results in diminished species diversity, disrupted ecosystem processes, and fundamental changes to habitat structures (*Katsanevakis et al., 2014*). These environmental impacts have been extensively documented by researchers studying biological invasions and their consequences for natural systems (*Haubrock et al., 2024*). It has been hypothesized that global warming as result of anthropogenic activities, increases the spread rate of non-native species in the recipient environments, and also this spread is accelerated by broad environmental tolerance and/or the ability to rapidly adapt to new conditions (*Marras et al., 2015*; *Mavruk et al., 2024*; *Haubrock et al., 2024*; *Raffalli et al., 2024*).

A recent inventory reported that Turkish marine ichthyofauna consists of 552 species (*Bilecenoğlu, 2024*). In terms of number of species, the eastern coasts of Türkiye had the highest diversity with 477 species, followed by the Aegean Sea (466 spp.), Sea of Marmara (277 spp.), and Black Sea (162 spp.). *Bilecenoğlu (2024)* reported that the number of non-native fish species occurring in the Turkish seas has reached 90 species, corresponding to 16.3% of the total ichthyofauna.

The geographical barrier between the Mediterranean and the Red Sea was eliminated by the completion of the Suez Canal in 1869, and Mediterranean biodiversity and ecosystem functions started to be damaged and altered by the introduction of alien species (*Chrairi et al., 2024*; *Pichot et al., 2024*). These significant changes mainly occur between biotic factors such as prey-predator interactions and food competition between the native and the non-native species (*Gülşahin & Soykan, 2017*). Competition between non-native Randall's threadfin bream *N. randalli* and native Common pandora *P. erythrinus*, which are widespread in the Mediterranean region, has been already reported in the literature (*Yapici & Filiz, 2019*).

The fish species *Nemipterus randalli*, originally inhabits the western Indian Ocean region, with its native distribution encompassing India's coastal waters, the Persian Gulf, and extending from the Red Sea down to Madagascar's waters (*Russell, 1990*). This invasive species was first documented in the eastern Mediterranean Basin (*Golani & Sonin, 2006*), and within a twenty-year period, it has successfully expanded its range to reach the northern waters of the Aegean Sea along Türkiye's coastline (*Aydın et al., 2022*).

Blood comprises approximately 1.3–7% of a fish's total body weight and serves as one of the body's most dynamic components. The blood system, along with blood-forming organs, plays a crucial role in metabolic functions by facilitating gas exchange between organisms and their surrounding environment (*Fazio et al., 2013a*). Understanding blood biochemistry parameters provides valuable insights that researchers can utilize as sensitive indicators to evaluate physiological alterations, assess health status, and identify potential

toxicological effects in fish populations (*Tavares-Dias, 2006*; *Sayed, Mekkawy & Mahmoud, 2011*; *Satheeshkumar et al., 2011*; *Satheeshkumar et al., 2012*).

Haematological and biochemical parameters can reveal important information about the fish physiology and different feeding habits and they are important tools to evaluate the fish health status (*Parrino et al., 2018*; *Acar et al., 2019*; *Fazio, 2019*). They can also be used to identify systematic relationships between specific species and to assess the adaptation of fish to the environment (*Pandey, 1977*; *Pavlidis et al., 2007*). However, one of the difficulties in assessing the health of fish species in the natural habitats is the scarcity of reliable references.

This study aims to determine and compare the reference values of the haematological and biochemical blood parameters of *Nemipterus randalli* (invasive) and *Pagellus erythrinus* (native) species, which live in the same habitat and compete, obtained from the Gökova coast of Muğla province of Türkiye.

## MATERIALS & METHODS

### Fish sampling

A total of 100 fish, 50 *N. randalli* (body weight 75.57 ± 7.89 g) and 50 *P. erythrinus* (body weight 69.38 ± 8.03 g), were collected from ten different stations in Gökova Bay (Muğla, Türkiye) between February 2023 and July 2024 at depths of 30–60 m in Gökova Bay (Fig. 1). Thus, samplings were conducted during the summer and winter seasons when water temperatures vary significantly.

In order to ensure that the fish were obtained alive and blood could be taken, fishing line sampling method was used. All fish were obtained from similar depths by the same method and anaesthetized to reduce stress. The sampled fish were visually examined and those without signs of disease were considered healthy and blood sampling was performed from these fish.

Before blood sampling, fish were anesthetized in a tank using 2-phenoxyethanol at a concentration of 200 mg/L (*Barata et al., 2016*). Blood samples were collected *via* caudal vein with a sterile plastic syringe (2.5 mL) and blood samples were transferred to two different tubes, one containing EDTA and the other without anticoagulant. The time between capture and blood collection was less than 5 min. After blood sampling, fish were weighed *via* using an analytical balance with 0.01 g precision. The collected blood was centrifuged at 4000 rpm for 10 min to separate serum for biochemical analyses.

During the fish samplings, the main physico-chemical water quality parameters (temperature, dissolved oxygen, salinity and pH) were also measured by using a portable multi-parameter device (YSI Professional Plus) and water quality parameters are given in Table 1.

All applications in the study are conducted in accordance with the standards recommended by the Guide for the Care and Use of Laboratory Animals and Directive 2010/63/EU for animal experiments and approved by MSKU-HADYEK Local Ethics Commission of Animal Research (Approval No: 2022/5-1).

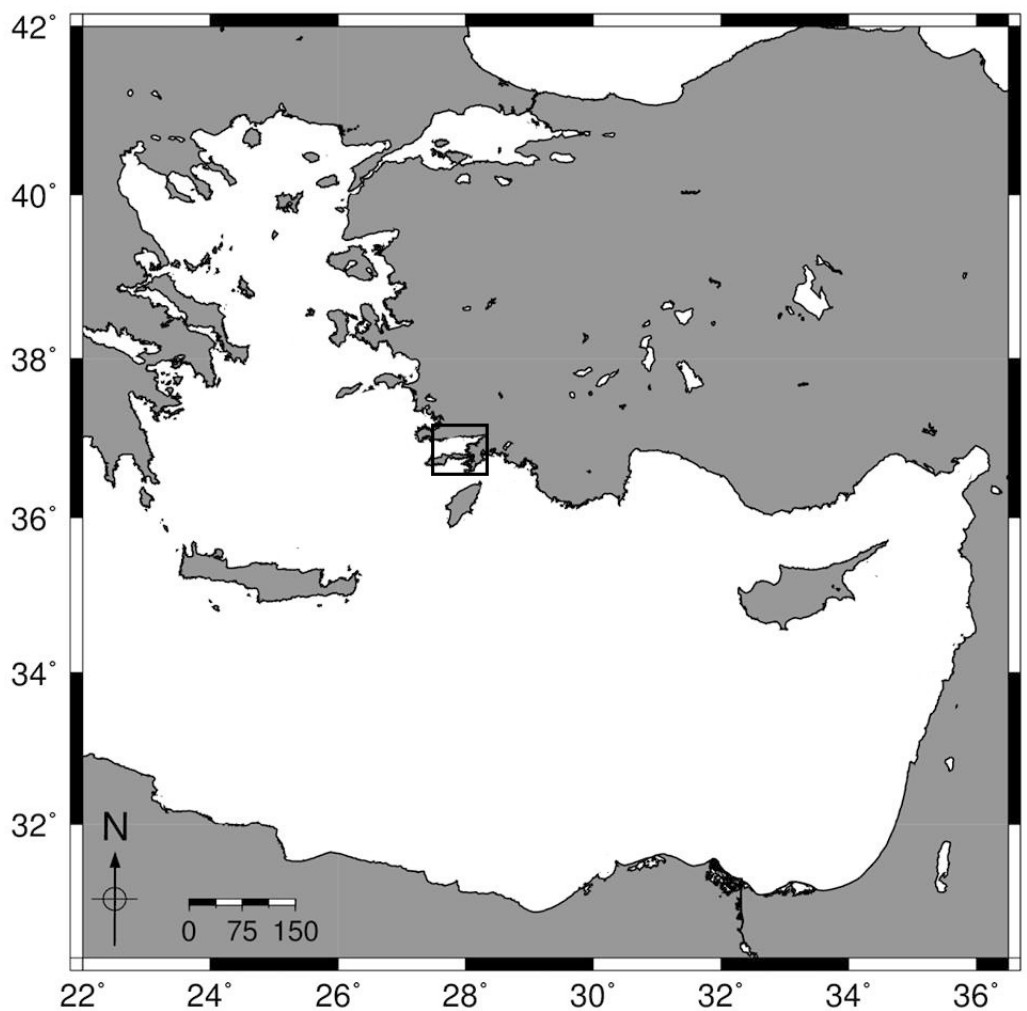

**Figure 1** Gökova Bay, the study area.

**Table 1 Physico chemical water quality parameters of water.**

| Parameters | Temperature (°C) | Dissolved oxygen (mg/L) | Salinity (‰) | (pH) |
|---|---|---|---|---|
| Winter | 17,5 ± 0,3 | 7,84 ± 0,17 | 35,50 ± 0,7 | 8,24 ± 0,1 |
| Summer | 23,6 ± 0,6 | 6,83 ± 0,17 | 37,25 ± 0,4 | 8,15 ± 0,1 |

## Haematological and biochemical analysis

Blood biochemical parameters were analyzed *via* using commercial test kits (Bioanalytic Diagnostic Industry, Co.). The analysis included key metabolic indicators such as Glucose (GLU), triglyceride (TRIG), cholesterol (CHOL), and protein components including serum total protein (TP), albumin (ALB), and globulins (GLO). Additionally, enzyme activities included aspartate aminotransferase (AST), alanine aminotransferase (ALT), alkaline

phosphatase (ALP) and lactate dehydrogenase (LDH) were measured. All measurements were conducted using a Shimadzu spectrophotometer (PG Instruments, UK).

Red blood cells, haematocrit and haemoglobin concentration were determined according to the method of *Blaxhall & Daisley (1973)*. Red blood cells (RBC) were counted with a Thoma haemocytometer by using Dacie's diluent. Haemoglobin concentration was measured using a spectrophotometer (540 nm) with the cyanometheamoglobin method. Mean erythrocyte volume (MCV), mean haemoglobin per erythrocyte (MCH) and mean haemoglobin concentration per erythrocyte (MCHC) which are called erythrocyte indices, were calculated using the following formulas.

MCV (fl) = Hct × 10/RBC ($10^6$/μL)

MCH (pg/cell) = [Hb (g/dL) × 10]/RBC($10^6$/μL)

MCHC (%) = [Hb (g/dL) × 100]/Hct.

### Statistical analysis

The values obtained were determined as mean ± standard deviation (SD). Kolmogorov–Smirnov normality test and Levene's test were used to analyse the normality and homogeneity of variance of the data, respectively. Differences in the haematological and biochemical parameters between the groups were analysed statistically by using one-way analysis of variance (ANOVA). The differences between means were determined by the Post-Hoc Tukey test and Tamhane's T2 test and a value of 0.05 was assumed as significant for *p* value. A multidimensional scaling (MDS) was applied to evaluate differences and similarities between the groups in terms of blood parameters. MDS based on proximity matrix resulted from Euclidean distance and z score values are used to standardize the data's. All data were analyzed by using the software SPSS version 22.

## RESULTS

The means and standard deviations of blood parameters of *N. randalli* and *P. erythrinus* species are given in Tables 2 and 3. In the study, it was observed that RBC, Hb and Hct values were higher in summer than in winter in both species. In both summer and winter sampling, it was observed that *N. randalli* had higher values than *P. erythrinus* for all three parameters and there was statistically significantly difference between the species.

The highest of MCV and MCH values were observed in *P. erythrinus* in both summer and winter. There was a significant difference between the two species MCV and MCH values in both summer and winter. The highest MCHC values were observed in *N. randalli* and there was a significant difference between the two species in winter but there is no significant difference in summer.

There was no statistically significant difference between the two species in terms of serum biochemical parameters GLU, TRIG, CHOL, TP, AST and ALP values in both summer and winter samples. However, ALT levels were measured in *N. randalli* higher than *P. erythrinus* and there was a significant statistical difference between the two species in both summer and winter samples. Additionally, the highest LDH value was obtained from *N. randalli* in the summer sampling and this value had a statistically significant difference from others.

**Table 2  Statistical results for the evaluated haematological parameters in *N. randalli* and *P. erythrinus* species.**

| Parameters | Species * sampling time | | | | P value |
|---|---|---|---|---|---|
| | *P. Erythrinus* | | *N. Randalli* | | |
| | *(Summer)* | *(Winter)* | *(Summer)* | *(Winter)* | |
| RBC ($\times 10^6$/μL) | $4,21 \pm 0,54^c$ | $2,00 \pm 0,48^a$ | $5,12 \pm 0,54^d$ | $3,37 \pm 0,61^b$ | 0,000 |
| Hb (g/dL) | $10,03 \pm 0,86^c$ | $6,18 \pm 0,84^a$ | $11,02 \pm 0,85^d$ | $8,05 \pm 0,99^b$ | 0,000 |
| Hct (%) | $38,38 \pm 2,10^b$ | $33,92 \pm 1,80^a$ | $41,62 \pm 2,16^c$ | $38,62 \pm 2,14^b$ | 0,000 |
| MCV (fL) | $92,17 \pm 9,56^b$ | $134,31 \pm 17,78^d$ | $81,82 \pm 6,26^a$ | $104,62 \pm 11,27^c$ | 0,000 |
| MCH (pg/cell) | $24,04 \pm 2,53^{bc}$ | $26,18 \pm 3,76^c$ | $21,64 \pm 1,88^a$ | $23,22 \pm 2,79^{ab}$ | 0,000 |
| MCHC (%) | $26,12 \pm 1,50^c$ | $19,24 \pm 1,42^a$ | $26,46 \pm 1,14^c$ | $22,08 \pm 1,33^b$ | 0,000 |

**Notes.**
Values with different lowercase letters in the same row indicate significant differences between the groups ($P < 0.05$).
RBC, red blood cells; Hb, haemoglobin concentration; Hct, haematocrit; MCV, mean corpuscular volume; MCH, mean corpuscular haemoglobin; MCHC, mean corpuscular haemoglobin concentration.

**Table 3  Statistical results for the evaluated biochemical parameters in *N. randalli* and *P. erythrinus* species.**

| Parameters | Species * sampling time | | | | P value |
|---|---|---|---|---|---|
| | *P. Erythrinus* | | *N. Randalli* | | |
| | *(Summer)* | *(Winter)* | *(Summer)* | *(Winter)* | |
| GLU (mmol/L) | $94,6 \pm 16,3^c$ | $75,7 \pm 15,48^{ab}$ | $86,5 \pm 14,3^{bc}$ | $73,0 \pm 12,1^a$ | 0,000 |
| TRIG (mmol/L) | $48,6 \pm 11,9^b$ | $36,0 \pm 11,5^a$ | $49,0 \pm 16,4^b$ | $37,5 \pm 12,8^a$ | 0,000 |
| CHOL (mmol/L) | $179,1 \pm 23,6^a$ | $174,9 \pm 28,5^a$ | $181,7 \pm 28,6^a$ | $172,6 \pm 27,2^a$ | 0,635 |
| TP (g/L) | $22,9 \pm 1,6^b$ | $20,1 \pm 2,8^a$ | $22,7 \pm 2,2^b$ | $18,7 \pm 1,7^a$ | 0,000 |
| AST (U/L) | $33,6 \pm 17,0^{ab}$ | $27,2 \pm 13,4^a$ | $38,5 \pm 15,6^b$ | $34,0 \pm 12,6^{ab}$ | 0,067 |
| ALT (U/L) | $2,6 \pm 1,2^a$ | $2,7 \pm 0,9^a$ | $3,8 \pm 0,9^b$ | $3,7 \pm 1,2^b$ | 0,000 |
| ALP (U/L) | $17,8 \pm 5,8^a$ | $18,6 \pm 5,2^{ab}$ | $22,6 \pm 7,3^{ab}$ | $22,8 \pm 7,7^b$ | 0,010 |
| LDH (U/L) | $496,1 \pm 227,0^a$ | $392,3 \pm 179,2^a$ | $840,8 \pm 291,0^b$ | $485,6 \pm 175,8^a$ | 0,000 |

**Notes.**
Values with different lowercase letters in the same row indicate significant differences between the groups ($P < 0.05$).
GLU, glucose; TRIG, triglycerides; CHO, cholesterol; TP, total protein; ALB, albumin; GLO, globulins; AST, aspartate amino transferase; ALT, alanine aminotransferase; ALP, alkaline phosphatase; LDH, lactate dehydrogenase.

The MDS plot showed that *N. randalli* and *P. erythrinus* individuals were almost separated from each other, but the area that the groups occupy in the graph were almost equal (Fig. 2). The accumulation of differently coloured points on the graph shows a distinct separation within the two species. For instance, blood parameters of *P. erythrinus* and *N. randalli* sampled in winter and summer were generally grouped in different centers (Fig. 2). The winter samples (blue and orange) are generally located towards the right side, while the summer samples (green and red) are mostly located on the left side of the graph. For both species, a differentiation was observed between summer and winter samples. This shows that seasonal factors may cause differences in blood parameters.

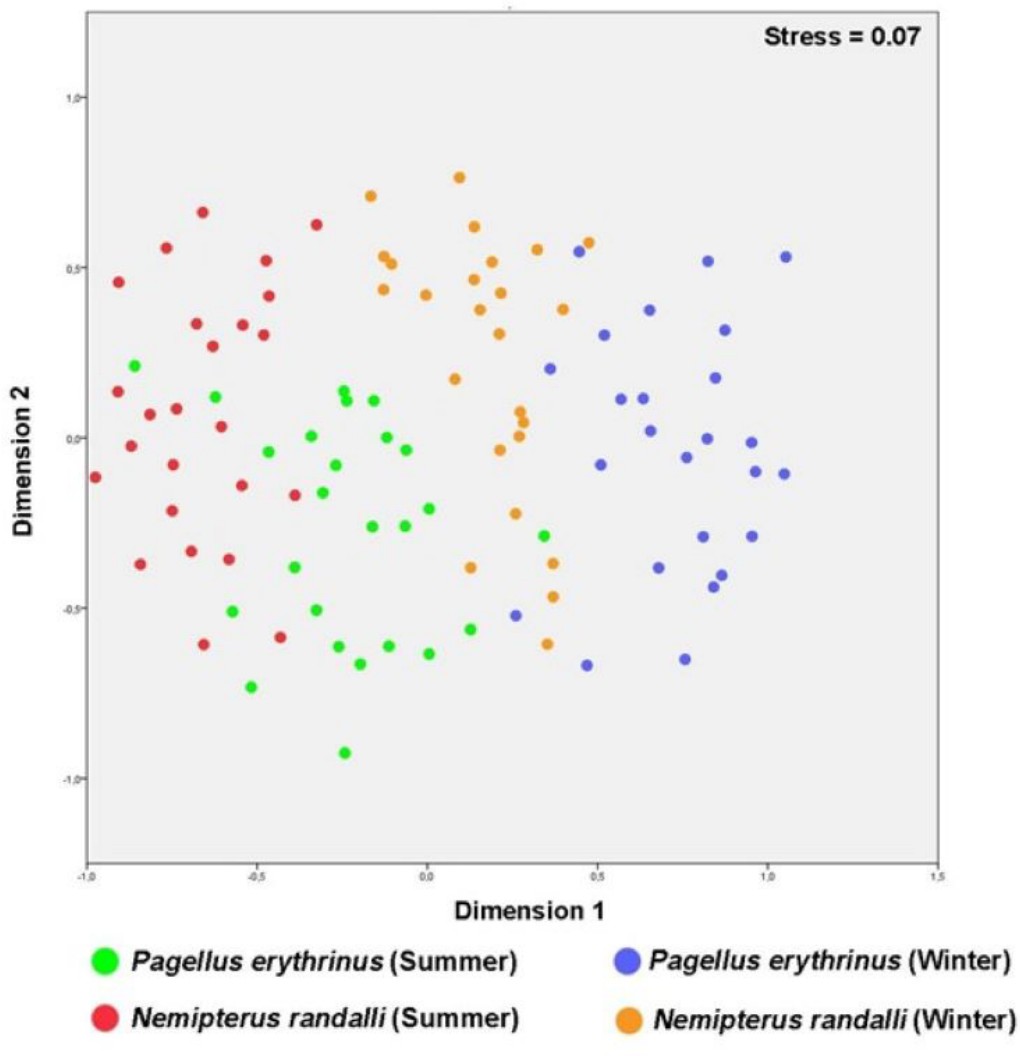

**Figure 2** Multidimensional scaling (MDS) plots of *P. erythrinus* and the *N. randalli* individuals based on blood parameters in summer and winter period.

## DISCUSSION

Blood parameters in fish, including both haematological and biochemical values, are subject to variation due to a broad range of factors. Internal factors that influence these parameters include the fish's size, age, genetic makeup, and reproductive stage. External influences encompass environmental conditions such as temperature fluctuations, photoperiod changes, and seasonal variations. Additionally, physiological factors like stress levels, nutritional status, and disease conditions can significantly affect blood values. Management-related aspects, including husbandry practices and water quality parameters, also play crucial roles in determining these biological markers (*Gabriel, Anyanwu & Akinrotimi, 2007*; *Vázquez & Guerrero, 2007*; *Yousefzadeh & Khara, 2015*). Additionally, comparative studies based on blood parameters are very important in better understanding

the systematic relationships among fish species. Such studies help to reveal physiological differences and similarities between species by comparing haematological and biochemical characteristics of different fish species. These data are not only useful to assess the health status of species, but also provide information on evolutionary affinities and adaptation strategies (*Fazio et al., 2013b*). In the present study, the changes in haematological and serum biochemical parameters of *Pagellus erythrinus* and *Nemipterus randalli* species spreading in Gökova Bay (Muğla, Türkiye) were determined in summer and winter seasons. In general, RBC, Hb and Hct values were found higher in summer than in winter for both species. According to *Svetina et al. (2002)*, the number of RBC increases at high temperatures. While comparing the RBC, Hb and Hct values the two species, the higher results were observed in *N. randalli* in both summer and winter seasons. This may be attributed to higher biometric parameters, but haematological parameter values are increase in active moving species and high RBC values are associated with fast-moving species (*Rambhaskar & Srinivasa Rao, 1987*; *Svobodova et al., 2008*). *Larsson, Johansson-Sjöbeck & Fänge (1976)* in their research on benthic species (*Lophius piscatorius* and *Cyclopterus lumpus*), which are less active than pelagic species, reported lower Hct and Hb values. Similarly, *Sayed, Mahmoud & Muhammad (2020)* in a study on *Parupeneus forsskali* and *Thalassoma klunzingeri* species found that RBC, Hb and Hct values were lower in *T. klunzingeri* which is a less active species. According to *Nikinmaa (2001)*, one way for fish to adapt to variable dissolved oxygen levels is through changes in the proportions of various hemoglobin's with different oxygen affinities. Another factor that increases the oxygen affinity of haemoglobin in fish is the adrenergic swelling of erythrocytes (*Lecklin, Tuominen & Nikinmaa, 2000*). The increase in the affinity of haemoglobin for oxygen with the decrease in MCHC indicates this mechanism is an adaptive response of *N. randalli* to the increase in oxygen demand (*Nikinmaa, 2001*).

When serum biochemical parameters of *P. erythrinus* and *N. randalli* are compared, remarkable differences are revealed regarding biochemical markers such as glucose (GLU), triglycerides(TRIG), cholesterol (CHOL), total protein (TP), aspartate aminotransferase (AST), alanine aminotransferase (ALT), alkaline phosphatase (ALP) and lactate dehydrogenase (LDH), depending on the species and sampling time. In both species, glucose levels fluctuate significantly depending on the seasons. Glucose values in *P. erythrinus* were higher in summer ($94.59 \pm 16.26$ mmol/L) and lower in winter ($75.73 \pm 15.48$ mmol/L), which may be arise due to the increased metabolic activity in warmer weather conditions. In other teleost species, higher glucose levels have been found to be associated with increases in metabolic demand (*Satheeshkumar et al., 2012*). *N. randalli* shows a similar pattern, with higher glucose levels in summer ($86.52 \pm 14.33$ mmol/L) and lower glucose levels in winter ($72.96 \pm 12.13$ mmol/L). These seasonal changes may reflect metabolic adjustments of species to adapt to environmental factors including water temperature and activity levels (*Strange, 1980*). A similar seasonal trend was seen in triglycerides, with higher levels observed in summer sampling in both species. These variations may indicate an increase in lipid metabolism due to fish reproductive activity or increased energy storage and use for physical effort in warmer waters (*Makri et al., 2024*). Cholesterol levels remained relatively constant for both species and no statistically significant difference was found

depending on the season and species. This indicates less seasonal effects on cholesterol levels. This stability of cholesterol reflects its role in maintaining cellular integrity and put forth that it may have a fundamental structural function rather than being a metabolic energy source (*Maxfield & Tabas, 2005*).  The absence of differences in glucose, cholesterol and triglyceride levels in *P. erythrinus* and *N. randalli* may indicate that these fish species haves similar feeding habits. Comparable glucose levels indicate similar energy metabolism, while stable cholesterol and triglyceride levels further support the idea that both species maintain similar lipid metabolism characteristics (*Coz-Rakovac et al., 2005*; *Hrubec, Smith & Robertson, 2001*). In both species, higher total protein concentrations were obtained in summer season. This may be related with the increased nutrition and metabolic activity during the summer season, which may have increased growth and protein synthesis. The decline observed during winter season may be due to reduced feeding activity and metabolic slowdown (*Fazio et al., 2012*). Liver enzymatic activities in fish are a critical indicator of liver function and overall metabolic health (*Lavanya et al., 2011*). Although AST levels showed a slight increase in summer in both species, this increase was not statistically significant. Although there were differences in ALT and ALP values among species, no seasonal differences were observed in both species. This suggests that liver activity may be similarly affected by environmental factors in both species (*Samanta et al., 2015*; *Zheng et al., 2022*). LDH, as an indicator of anaerobic glycolysis, shows significant differences between species in summer season (*Martínez et al., 2011*). *N. randalli* had significantly higher LDH levels in summer (840.75 ± 291.01 U/L) compared to winter (485.62 ± 175.82 U/L). This may be related with higher anaerobic metabolism or greater environmental stress. *P. erythrinus* showed relatively lower and less changing LDH levels, probably reflecting lower use of anaerobic metabolism (*Dando, 1969*).

## CONCLUSIONS

In conclusion, the haematological and biochemical parameters of *Pagellus erythrinus* and *Nemipterus randalli* species show significant seasonal changes, influenced by many internal and external factors such as temperature, feeding behaviors and environmental stressors. The increased erythrocyte, haemoglobin, and haematocrit levels observed during the summer season indicates that increased metabolic activity and growth due to increased nutrient availability and warmer water temperatures. Furthermore, this study revealed differences in enzymatic activities and biochemical markers between the two species. However, considering the seasonal changes in AST, ALT and ALP enzymes in both species, it is shown that both species give similar responses to changes in environmental factors. Differently, LDH value shows a more significant change in *N. randalli* species. This may be related to high anaerobic metabolic activity or environmental stress. In the study, considering the biological characteristics and origin of the *N. randalli* species, it could be expected that *N. randalli* could give more respond than the *P. erythrinus* to the environmental factors in the winter season. However, it was seen that both species give similar responses in all parameters except LDH. Contrary to expectations, this difference in LDH values occurred in the summer season, not in the winter season. This similarity

in haematological and biochemical parameters revealed the possibility that *N. randalli* could distribute to new habitats where *P. erythrinus* is distributed. It shows that the future distribution of *N. randalli*, an invasive species, should be carefully monitored.

Overall, comparative analysis of haematological and biochemical parameters provides valuable information about the physiological adaptations of these fish species to environmental conditions. These findings highlight the importance of monitoring such parameters to assess the health status and ecological status of marine fish. Further studies focusing on thoroughly investigating the impact of climate change on physiological traits as well as fish health and biodiversity are encouraged.

## ACKNOWLEDGEMENTS

The authors would like to thank to Prof. Dr. Daniela GIANNETTO for language editing.

### Funding

This study has been granted by the Çanakkale Onsekiz Mart University Research Projects Coordination Office through Project Grant Number: (FHD-2023-4278). The funders had no role in study design, data collection and analysis, decision to publish, or preparation of the manuscript.

### Grant Disclosures

The following grant information was disclosed by the authors:
Çanakkale Onsekiz Mart University Research Projects Coordination Office: FHD-2023-4278.

### Competing Interests

Sercan Yapıcı is an Academic Editor for PeerJ.

### Author Contributions

- Rifat Tezel conceived and designed the experiments, performed the experiments, analyzed the data, prepared figures and/or tables, authored or reviewed drafts of the article, and approved the final draft.
- Ümit Acar conceived and designed the experiments, performed the experiments, analyzed the data, prepared figures and/or tables, authored or reviewed drafts of the article, and approved the final draft.
- Sercan Yapıcı conceived and designed the experiments, performed the experiments, analyzed the data, prepared figures and/or tables, authored or reviewed drafts of the article, and approved the final draft.

### Animal Ethics

The following information was supplied relating to ethical approvals (*i.e.*, approving body and any reference numbers):

All applications in the study are conducted in accordance with the standards recommended by the Guide for the Care and Use of Laboratory Animals and Directive 2010/63/EU for animal experiments and approved by MSKU-HADYEK Local Ethics Commission of Animal Research (Approval No: 2022/5-1).

## Data Availability

The raw data is available in the Supplemental File.

## Supplemental Information

Supplemental information for this article can be found online at http://dx.doi.org/10.7717/peerj.18784#supplemental-information.

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
