# Peer review of "A comparative evaluation of haematological and biochemical parameters of Nemipterus randalli and Pagellus erythrinus species living in Gökova Bay, Türkiye"

_PeerJ, doi:10.7717/peerj.18784_

## Round 0.1 · original submission · Minor Revisions

Dear Author,

Thanks for your submission to PeerJ. "Minor Revision" was decided in line with the reviewer's feedback. Please revise your publication by considering the reviewer's report. All the best.

Servet

Reviewer 1 ·

Basic reporting

The manuscript “A comparative evaluation of haematological and biochemical parameters of Nemipterus randalli and Pagellus erythrinus species living in Gökova Bay, Türkiye” is an interesting work which focuses on evaluating the differences in haematological and biochemical blood parameters in two species, the Randall’s threadfin bream and the Common pandora.
Aim of the work is to detect the physiological adaptations of both the forementioned species, which are respectively an autochthonous and an allochthonous and invasive one, to the specific environment of the Gokova Bay, in order to rule out the possibility of a potential wider distribution of the allochthonous species to other environments.
The manuscript is written in a generally good English and is easy to follow; the research appears to be thoroughly carried out, with interesting results.
The Introduction is complete and provides good bibliographical notes.
Materials and methods are well structured, and the specifics of the research are clearly explained.
Results are complete and widely explained as well.
The Discussion is broad and provides sufficient bibliography, giving a complete overall perspective on the matter.
The Conclusions are coherent with the findings of the research.
However, it is suggested to look up at some typos which are listed below.
In line 27, please correct “are significant increases” with “significantly increare”.
In line 62, correct “have reached” with “has reached”.
In line 71, “Russell, 1986” either needs parenthesis or has to be deleted.
In line 92, please correct with “which live in the same habitat and compete”.
In line 173, please delete “there is”.
In line 179, please change “a diverse” with “a broad”.
In line 198, please correct with “increase”.
In line 205, please add “to” adapt.
In line 216, please correct “was” with “were”.
In line 225, please correct “increased” with “an increase”.
In my opinion, the present work can be accepted for publishing after a minor revision.

Experimental design

The experimental design was well organized, the fish were sampled and treated correctly and the haematological analyses were performed with scientific rigor and with methods applicable to fish.

Validity of the findings

Climate change and its impact on aquatic organisms is an interesting aspect and represents a valuable contribution for investigations using bioindicators. However, a significant contribution is needed to identify reference values ​​that are difficult to obtain in fish.

Additional comments

As the authors report, this is a preliminary study that requires further research. The interesting aspect of the study is the impact of the aquatic environment on the blood parameters of fish. Fish live in close contact with water and this determines important variations in blood parameters.

·

Basic reporting

I have presented all my opinions in section 4.

Experimental design

I have presented all my opinions in section 4.

Validity of the findings

I have presented all my opinions in section 4.

Additional comments

In this study, the authors compared the hematological and biochemical parameters of two fish species, Nemipterus randalli and Pagellus erythrinus, from Gökova Bay, Türkiye. This study highlights the differences in blood parameters between these two species, especially significant changes in red blood cell counts, hemoglobin, and hematocrit levels during summer.

Both species showed higher red blood cell (RBC), hemoglobin (Hb), and hematocrit (Hct) values ​​in summer compared to winter. This trend is understood as triggering increased metabolic activity and growth due to warmer water temperatures and greater nutrient availability.

In the comparative assessment of the two species, N. randalli showed consistently higher RBC, Hb, and Hct values ​​in both summer and winter seasons, which indicates the species' physiological adaptations and different responses to environmental conditions.

The authors underlined that various factors such as temperature fluctuations, stress levels, and nutritional status significantly affect the hematological parameters of these fish species and the importance of the findings in monitoring these parameters to assess the health and ecological status of marine fish.

Overall, the study provides valuable information about the physiological differences between the two species and their adaptations to seasonal changes in their environment to achieve a healthy marine ecosystem.

As a result of all these evaluations, I consider the article acceptable for publication.

Reviewer 3 ·

Basic reporting

Title: accepted
Abstract:
may provide some of the haematological and biochemical profiles values
any reason of selected the specific months for sampling?

Introduction

overall is accepted. However, there are two concerns:
1. Perhaps separate into several paragraphs
2. too many old references indicating the study is not up to date

Materials and methods

any references to support method in the line 99, 106

Experimental design

accepted

Validity of the findings

the findings can be a basic information

Annotated reviews are not available for download in order to protect the identity of reviewers who chose to remain anonymous.

---

## Round 0.2 · accepted · Accept

Dear Authors,

Thank you for your submission to PeerJ. Your MS in its current form can then be accepted.

Thank you for making the necessary arrangements.

Best regards Servet